# Occurrence and antimicrobial susceptibility of *Staphylococcus aureus* in dairy farms and personnel in selected towns of West Shewa Zone, Oromia, Ethiopia

**Milsan Getu Banu**[1], **Endrias Zewdu Geberemedhin**[2]*

**1** Chelia agricultural office, West Shewa Zone, Oromia, Ethiopia, **2** Department of Veterinary Science, School of Veterinary Medicine, Ambo University, Ambo, Ethiopia

* endrias.zewdu@gmail.com, endrias.zewdu@ambou.edu.et

**Data Availability Statement:** All relevant data are within the paper and its Supporting Information files.

## Abstract

*Staphylococcus aureus is one of the foodborne disease-causing bacterial pathogens. A cross-sectional study was conducted in selected towns of the West Shewa Zone, Oromia, Ethiopia from December 2020 to April 2021. The objectives of the study were to estimate the occurrence and load of S. aureus in raw cows' milk, the antimicrobial susceptibility patterns of the S. aureus isolates, and assess the knowledge, attitude, and practice of the farmers on factors of antimicrobial resistance. A total of 311 samples from raw cows' milk (212), milkers' hands (44), and milking buckets (55) swabs were collected and tested. The disc diffusion method was used to test the antimicrobial susceptibility of the isolates. A questionnaire survey was conducted to assess the factors of milk contamination with S. aureus and antimicrobial resistance. The Chi-square test, one-way analysis of variance, and logistic regression analysis were used for data analyses. The result indicated that 16.72% (52/311) (95% CI: 12.75–21.34%) of the samples were positive for S. aureus. The occurrence of S. aureus was 22.73%, 16.51%, and 12.73% in milkers' hand swabs, cow milk, and milking bucket swabs, respectively. The mean count of S. aureus from raw cows' milk was 4.3± 1.45 log10 CFU/ml. About 88% of S aureus isolates were resistant to ampicillin while 82.9% and 70.7% of the isolates were susceptible to ciprofloxacin and cefotaxime respectively. The majority of the S. aureus isolates (61%) showed multi-drug resistance. The odds of S. aureus isolation from the milk of cows were significantly high in older cows (Adjusted Odds Ratio [AOR]: 5.54; p = 0.001), in late lactation stages (AOR: 3.6; p = 0.012), and in farms where house cleaning was done twice per week (AOR: 8.7; p = 0.001). A high percentage of farmers had insufficient knowledge, attitude, and practice (KAP) about the factors contributing to antimicrobial resistance. In conclusion, the poor milk hygienic practices, high rate of antimicrobial resistance (AMR), and inadequate KAP of farmers about factors of AMR suggest potential public health risks thus requiring training and surveillance programs.*

**Funding:** The author(s) received no specific funding for this work.

**Competing interests:** The authors have declared that no competing interests exist.

## 1. Introduction

*Staphylococcus aureus* (*S. aureus)* is an opportunistic bacterial pathogen that can survive and multiply in a range of conditions. It is frequently blamed for being one of the major causes of foodborne diseases (FBDs) [1, 2]. It is recognized as the third most reported FBD-causing pathogen in the world [3]. In addition to FBDs, *S. aureus* results in public health problems and a large financial loss in the dairy industry (associated with mortality, culling of infected dairy cows, spoiling of the milk, lower shelf life, decreased yield of milk products, cost of treatment, decreased milk quality, and loss of the milk due to drug residue) [4, 5]. Recently, *S. aureus is also* incredibly adaptive and continuously evolves resistance to most of the available antimicrobials including the emergence of methicillin-resistant *S. aureus* (MRSA) strain which poses a severe challenge to both veterinary and human health around the globe [6]. It is also liable for the contamination of food products leading to food spoilage, reduction of food safety, and shelf life [7].

Staphylococcal food poisoning (SFP) is also another cause of FBDs in many parts of the world which results from the consumption of food contaminated with a sufficient amount of pre-formed *Staphylococcal* enterotoxins (SEs) [8]. The outbreaks of FBDs associated with dairy and dairy products in developing and industrialized countries are 1–10% and 2–6% respectively [9]. In developing nations including Ethiopia, controlling the hygiene of food is very low and it is largely approved for human consumption based on visual inspection. These actions can promote foodborne and zoonotic pathogen transmission to humans [10]. In Ethiopia, there is gradual growth and expansion of small and medium-scale dairy farms in urban and peri-urban areas and the demand for milk is also increasing due to urbanization, a hastily growing population, and an increasing preference toward animal-sourced food [11]. The consumption of milk in Ethiopia was reported to be between 17 and 19 liters per capita, which is lower than the regional average (for instance, the amount of milk consumed per capita in Uganda and Kenya is 50 and 90 liters, respectively) [12]. The production of milk in the country is lower than the requirement, and this might be due to versatile factors; including diseases, improper production of dairy animals, and milk processing practices in farms [13].

The load and public health effects of FBDs associated with *S. aureus* are poorly understood in Ethiopia. But, the epidemiology of this microorganism and the significant raw milk consumption habits in the population are suggestive of acquiring *S. aureus* [14]. The consumption of raw milk is common in Ethiopia and most people typically prefer raw milk because of its flavor, availability, and price [12]. However, it isn't always safe for a consumer from a health point of view due to the production often taking place in unsatisfactory hygienic situations [15]. As a result, there is a high probability of the occurrence of SFP because of the consumption of dairy products. The burden of *S. aureus* is a major issue in figuring out milk quality and hygienic levels exercised during milking [16]. In Ethiopia, research done so far indicated 14.9% occurrence of *S. aureus* in dairy farms samples at Asella town [17], 32% from milker's hands at Sebeta town [15], 16.6% from dairy farms at Sululta and Mukaturi towns [18], and 17% from dairy farms at Mekelle [19]. There is only one report on the prevalence and risk factors of mastitis and antibiogram of *Staphylococcus* species from mastitis-positive Zebu (*Bos indicus*) cows in the Cheliya and Dendi districts of West Shewa Zone [20]. Additionally, in the Ejere district, there is only a report on the safety and quality of raw whole cow milk produced and marketed by smallholders [21].

However, there is a limitation of data regarding the prevalence of *S. aureus*, potential sources of contamination of milk and handling practices, the load of *S. aureus* in raw cow milk, and antimicrobial susceptibility of *S. aureus* isolated from milk, milker's hand, and milking utensil in the current study areas. Furthermore, there is no data on the knowledge,

attitudes, and practices (KAP) of farmers concerning factors contributing to the occurrence of antimicrobial resistance (AMR) both in humans and animals at the farm level, which might help to formulate strategies to maximize and maintain the benefits of antimicrobial use (AMU) in livestock production with minimal effects to human health. Therefore, the study aimed to estimate the occurrence of *S. aureus* in raw milk, milking equipment, and personnel working on dairy farms, enumerate *S. aureus* in cow's milk, determine the antimicrobial susceptibility patterns of the *S. aureus* isolates, and assess the KAP of the dairy farmer's/farm workers on the factors contributing to antimicrobial resistance in dairy cows and personnel.

## 2. Materials and methods

The study was conducted in three towns (Ejere, Ginchi, and Gedo) of West Shewa Zone Oromia Regional State, Ethiopia. These towns were purposively selected due to accessibility, milk production potential, and the absence of similar previous studies. The Ejere, Ginchi, and Gedo towns are located at a distance of 40 km, 75 km, and 178 km West of the Ethiopian capital city, Addis Ababa, along Nekemte road respectively. The location, altitude, rainfall, temperature, and human and cattle population of these towns are depicted in Table 1.

### 2.1. Study design

A cross-sectional study design was conducted from December 2020 to April 2021 to estimate the occurrence and antimicrobial susceptibility of *S. aureus* in dairy farms and personnel. The samples were collected from randomly selected small and medium-scale dairy farms.

### 2.2. Study populations and animals

The study population of this research was apparently healthy crossbreed (Zebu with Jersey and Holstein-Friesians) lactating cows of all age categories found in randomly selected dairy farms (small and medium scale) kept under intensive and semi-intensive management systems in the study area. A total of 74 farms consisting of 51 small and 23 medium-scale dairy farms were selected by simple random sampling techniques from 94 dairy farms found in the study towns. The study animals of this research were selected by the simple random sampling technique from cross-breed lactating cows in the selected farms.

### 2.3. Sample size determination

The sample size was calculated using the formula described by Thrusfield [25]. The considerations were a 95% confidence interval and 5% desired absolute precision. The required sample size was calculated considering a previously published prevalence estimate of *S. aureus* in dairy

**Table 1. Information about the study towns.**

| | | Study towns | | |
|---|---|---|---|---|
| | | **Ejere** | **Ginchi** | **Gedo** |
| Coordinate | Latitude | 9˚2 'N | 0 9˚ 01 'N | 9˚ 0′ 0″ N |
| | Longitude | 38˚24'E | 38˚ 10′E | 37˚-38˚E |
| | Altitude in meters above sea level | 2060–3185 | 2200 | 1700–3060 |
| Average of Annual | Rainfall in mm | 1100 | 1140 | 750–1000 |
| | Temperature in˚C | 26. 5 | 16.3 | 16 |
| Population | Human | 114,714 | 145,129 | 108,396 |
| | Cattle | 116,685 | 192,345 | 85,443 |
| Reference | | [22] | [23] | [24] |

farms (16.6%), reported by Regasa *et al*. [18] on milk safety assessment, isolation, and antimicrobial susceptibility profile of *S. aureus* in selected dairy farms of Mukaturi and Sululta towns, Oromia region, Ethiopia.

$$N = \frac{1.96^2 P \ exp[1 - Pexp]}{d^2}$$

Where; *N* = required sample size, *P* = expected prevalence, and *d* = desired absolute precision.

Therefore, the calculated sample size was 212, and the sample size was proportionally distributed in each town based on the number of dairy cows. Additionally, forty-four (44) swab samples from milkers' hands and fifty-five (55) swab samples from milking buckets were collected by purposive sampling technique. Overall, 311 samples were collected and subjected to microbiological testing.

## 2.4. Questionnaire survey

A pre-structured questionnaire survey was conducted to determine the status of hygienic activities in dairy farms such as house cleaning, udder cleaning, hand washing, and other conditions that were thought to influence the hygienic quality of raw cow milk. The KAPs of dairy farmers/workers on factors contributing to the occurrence of AMR in dairy cows and personnel was assessed through the pre-structured questionnaire survey. A total of 74 individual farm owners/farm workers from each study area were interviewed to generate data on milking hygiene, milk handling practice, housing, and cleaning practice on the farms. Types of farms were also classified as medium scale ($> 5 \leq 30$ dairy cows), and small scale ($\leq 5$ dairy cows) [26]. The farms were classified into intensive (if cows are managed under confinement with supplementation of feeding and watering) and sem-intensive (if animals are partly confined and allowed to graze freely or under paddocking, supplementation of diet in addition to natural pasture) [27].

Additionally, KAPs of the farmers or farm workers on the factors contributing to AMR in dairy cows and personnel were assessed at the farm level according to the following procedures. Under the knowledge, seven questions were included to assess the farm owners'/workers' knowledge of factors contributing to AMR. Of these seven questions, four were Likert scale, two were yes/no, and one open-ended question that allows the respondents to express their perceptions. Each close-ended question received three points for a correct answer, two points for an uncertain answer, and one point for a wrong answer. The score varied from 4–12 points and was classified into 3 levels according to Bloom's [28] cutoff point, 60–80% as follows; high level (80–100%) 10–12 scores, moderate level (60–79%) 7–9 scores, and low level ($< 60\%$) 4–6 scores.

In the attitude part, seven questions, with Likert scale options of choice ranging from 'agree' to 'disagree', were included. The scores varied from 7 to 21 and all individual answers were summed up for the total and calculated for means. The scores were classified into 3 levels (positive, neutral, and negative attitude). Positive attitude 17–21 scores (80–100%), neutral attitude 12–16 scores (60–79%), and negative attitude 7–11 scores ($< 60\%$).

In the practice part, ten questions were included. From these ten questions, eight Likert scales and two yes/no questions were encompassed. The rating scale of responses was measured in knowledge and attitudes. The scores in measuring the practice of antimicrobial usage varied from 8 to 24, and the levels of practice were; good (80–100%) 18–24 scores, fair (60–79%) 15–17 scores, and poor ($< 60\%$) 8–14 scores.

## 2.5. Inclusion and exclusion criteria

The inclusion criteria for this study were the availability of one or more cross-breed lactating cows at the time of sample collection, the willingness of dairy farm owners to provide milk, and the willingness of milkers to participate in the study and give the necessary information via questionnaire. The exclusion criteria of this study were local lactating cows (Zebu) since they aren't reared under intensive or semi-intensive management systems, those farmers who weren't around during the study, unwilling to participate in the study, farm owners with difficulty of hearing and unable to give the required information, and those who had no time for questionnaire interviews.

## 2.6. Sample collection and transportation

Before the collection of the milk sample, the udder of the cow and teats were cleaned and dried. Then each teat's end was scrubbed gently with cotton swabs moistened and disinfected with 70% ethyl alcohol before sampling. About 10 ml of raw milk was taken from four-quarters of each cow using a sterile test tube (corked) after discarding the first 3–4 streams of milk from the quarters [29]. Samples of hand swabs were taken by rubbing sterile cotton-tipped swabs onto the palms of both hands, the area between fingers and fingertips, and rotating the swab 360˚ between the bases of the fingers of both hands of the milkers before milking. The swabs of milking buckets were also taken by streaking an estimated area of 10 $cm^2$ on the surface of the inner wall of milking buckets; which are commonly used for milking. These swab samples were taken before milking [30]. The collected swab samples from the milker's hand and milking buckets were kept in a sterilized test tube containing 9 ml of buffered peptone water. The samples were then appropriately labeled and delivered to Ambo University's Laboratory for Zoonosis and Food Safety in a colder environment with the use of an icebox filled with ice packs. Upon arrival, the collected samples were immediately stored at 4˚C until culturing.

## 2.7. Isolation and identification of *Staphylococcus aureus*

The bacteriological culture was conducted according to standard microbiological techniques [29]. A loopful of milk and swab samples were streaked on sterile 5% defibrinated sheep blood agar by using an inoculating loop and the plates were incubated aerobically at 37˚C and examined after 24–48 hours of incubation. The colonies that cause haemolytic patterns on blood agar plates were cultured on nutrient agar and checked for colony characteristics (round, smooth, convex, and golden yellow coloured appearance). The representative colonies which were positive for Gram's staining and typical grape-like structure under the microscope were further sub-cultured on nutrient agar plates and incubated at 37˚C for 24 hours. The pure colonies were preserved and maintained on nutrient slants for further characterization of the isolates. Finally, *S. aureus* was identified based on biochemical tests such as catalase, Mannitol salt agar, Purple agar base, and Coagulase tests. The samples were found to be positive for *S. aureus* when the isolates are positive for catalase and coagulase and display fermentation of mannitol salt agar and maltose (strong yellow discoloration of both media) [29].

## 2.8. The enumeration and interpretation of *Staphylococcus aureus* from raw cow milk

Parallel to inoculation on blood agar, 10-fold serial dilutions of milk samples were prepared up to $10^{-6}$ in normal saline water [29]. Then, from each dilution, 0.1 ml of sample suspension was aseptically transferred to the Baird-Parker Agar (Sisco, India) plate. The colony-counting test was done for 112 randomly selected raw milk of the total milk samples. The plate containing

colonies with the typical appearance of circular, smooth, convex, moist, and gray to jet-black, frequently with a light-colored (off-white) margin, surrounded by an opaque zone with an outer clear zone in the medium was taken as *S. aureus*. Plates that contained 20–200 typical colonies were selected for the *S. aureus* count. However, when there is no typical colony of *S. aureus* found in the higher dilution; the plate with <20 and >200 typical colonies in the lower dilution was used [30]. Finally, the total *S. aureus* colonies counted from two consecutive plates of each sample was converted into colony-forming units per milliliter (CFU/ml) using the following formula [31].

$$N = \frac{\sum C}{V(n_1 + 0.1n_2)d}$$

Where; $N$ = number of bacterial colonies counted, $C$ = the sum of colonies identified on two consecutive dilution steps, where at least one contained 20 colonies and less than 200 colonies, $n_1$ is the number of plates counted at the first dilution, $n_2$ is the number of plates counted at the second dilution, $V$ = volume of inoculums on each *dish/plate* in milliliter, and $d$ = dilution factor corresponding to the first dilution selected (the initial suspension is a dilution).

## 2.9. Antimicrobial susceptibility testing

The antimicrobial susceptibility test of isolates was performed using the disc diffusion method on Muller-Hinton agar (HiMedia, India) plates as recommended by the National Committee for Clinical Laboratory Standards [32]. The isolates of *S. aureus* were subjected to 13 antimicrobial susceptibility-testing discs (Oxoid, UK). About 2–3 pure colonies of the isolates were taken from the nutrient agar (HiMedia, India) and suspended in a tube containing 5 ml of tryptose soya broth (HiMedia, India) and then, incubated at 37°C for 1–2 hours. The turbidity of the suspension was adjusted to the density of 0.5 McFarland standard (0.5 ml of 1% *w/v* $BaCl_2$ and 99.5 ml of 1% *v/v* $H_2SO_4$) of approximately $3 \times 10^8$ CFU/ml by adding a sterile saline solution or more colonies to standardize the size of the inoculum. A sterile cotton swab was dipped into the standardized suspension of the bacterial culture, squeezed against the sides of the test tube to remove the excess fluid, and inoculated into Mueller-Hinton agar (HiMedia, India) and the plates were held at room temperature for 15 minutes to allow drying of the flood. The discs were placed with a 20 mm gap between each other and 15 mm from the edge of the plates to prevent overlapping of the inhibition zones [32]. The known positive control used as a standard was *Staphylococcus aureus* ATCC 25923. The following fourteen antimicrobial discs, *i.e.* gentamycin (10μg), cefotaxime (30μg), ceftriaxone (5μg), azithromycin (30μg), nitrofurantoin (300μg), ciprofloxacin (5μg), norfloxacin (10μg), nalidixic acid (30μg), cotrimoxazole (25μg), tetracycline (30μg), ampicillin (10μg), oxacillin (1 μg), vancomycin (30μg), and chloramphenicol (30μg) were used for antimicrobial susceptibility testing. The plates were allowed to stand for 30 minutes for the diffusion of the active substance of the agents and incubated at 35–37°C for 24 hours in an upside-down position. Finally, the diameters of the zone of inhibition around the discs were measured to the nearest millimeter using a ruler, and the isolates were classified as susceptible (S), intermediate (I), and resistant (R) [32]. The antimicrobials were selected based on their availability during the research work and habitual uses in human and animal medications. The isolates showing resistance to three or more antimicrobial subclasses were considered multiple drug-resistant (MDR) [33].

## 2.10. Data management and analysis

All collected data were entered into the Microsoft Excel spreadsheet, cleaned, and verified before entering into the STATA Data Editor View (Version 14.2) for statistical analysis

(STATA Corporation College Station, TX, USA). Descriptive statistics were used to summarize data in tables and graphs. The Chi-square test was used to test the association of KAPs of the farmers with the factor contributing to the occurrence of AMR. The one-way ANOVA was used to test the mean count of *S. aureus* in the raw cow milk after the Log10 transformation of *S. aureus* count was done and the mean difference of MDR of *S. aureus* isolated from milk, hand swab, and milking buckets was also analyzed by one-way ANOVA. Univariable and multivariable logistic regression analyses were done and 95% confidence intervals (CI) were calculated for statistical significance tests for the factors leading to the occurrence of *S. aureus* in the milk. Non-collinear variables with a p-value $\leq 0.25$ in the univariable logistic regression analysis were considered for the multivariable analysis to look for a relative effect on the outcome variables by controlling other possible confounding factors and the level with the lowest prevalence of the risk factors was used as a reference category. Differences were considered statistically significant at $p < 0.05$.

## 2.11. Ethical approval

The Ambo University Research and Ethical Committee reviewed and approved this work. The International guidelines for the Care and Use of Laboratory Animals were strictly followed when conducting this investigation. The University of Ambo's Research and Ethical Committee examined the proposal for the animal part and granted a waiver of ethical approval.

# 3. Results

## 3.1. Prevalence of *Staphylococcus aureus*

Out of the total 311 samples tested in this study, the overall prevalence of *S. aureus* was found to be 16.72% (95% Confidence Interval [CI]: 12.75–21.34%). The prevalence of *S. aureus* in the raw cow milk, milker's hand, and milking bucket swabs was found to be 16.51%, 22.73%, and 12.73%, respectively (Table 2). The frequency and percentages of the isolation of *S. aureus* varied among towns and sample types. In this study, the prevalence of *S. aureus* was high in milker's hand swabs (22.73%) compared to milk (16.51%) and milking buckets (12.73%).

**Table 2. Percentage of *S. aureus* isolated from raw milk, hand, and milking buckets swab samples in the study areas.**

| Study area | Types of sample | Number positive/tested | % Prevalence (95% CI) |
|---|---|---|---|
| Ejere | Milk | 16/91 | 17.60 (9.53–25.73) |
| | Hand Swabs | 5/21 | 23.80 (8.22–47.20) |
| | Buckets Swabs | 4/26 | 15.40 (4.40–34.90) |
| Ginchi | Milk | 11/67 | 16.40 (8.50–27.50) |
| | Hand Swabs | 3/14 | 21.40 (4.70–50.80) |
| | Buckets Swab | 2/17 | 11.80 (1.50–36.40) |
| Gedo | Milk | 8/54 | 14.80 (6.20–27.10) |
| | Hand Swabs | 2/9 | 22.20 (2.80–60.00) |
| | Buckets Swabs | 1/12 | 8.30 (0.21–38.50) |
| Total of each sample | Milk | 35/212 | 16.51 (11.50–37.80) |
| | Hand swabs | 10/44 | 22.73 (10.30–35.10) |
| | Bucket swabs | 7/55 | 12.73 (5.30–27.10) |
| Overall prevalence | | 52/311 | (12.75–21.34) |

**Table 3. The factors contributing to the occurrence of *S. aureus* in raw milk in the study area.**

| Factors | Category | Number tested | Log10CFU/ml of *S. aureus* Mean | P-value |
|---|---|---|---|---|
| Study towns | Ejere | 46 | 3.61 | 0.291 |
| | Ginchi | 36 | 4.40 | |
| | Gedo | 30 | 4.63 | |
| Hand washing before milking | Yes | 57 | 4.17 | 0.284 |
| | No | 55 | 4.09 | |
| Hand washing b/n each milking | Yes | 41 | 3.95 | 0.930 |
| | No | 71 | 4.22 | |
| Udder cleaning intervals | Only before milking | 19 | 3.55 | 0.398 |
| | After and before milking | 32 | 3.78 | |
| | No cleaning | 61 | 4.43 | |
| Using drying towel separately | Yes | 6 | 2.20 | 0.970 |
| | No | 106 | 4.20 | |
| Drainage system | Yes | 71 | 3.98 | 0.405 |
| | No | 41 | 4.37 | |
| Total mean | | | 4.3 ±1.45 log10CFU/ml | |

### 3.2. The load of *Staphylococcus aureus* in raw cow milk

An overall mean counts of the *S. aureus* in raw cow milk of the Ejere, Ginchi, and Gedo towns were 3.61, 4.40, and 4.63 log10CFU/ml respectively, with the total mean and standard deviation of 4.3 ±1.45 log10 CFU/ml (Table 3). The one-way analysis of variance showed that there was no significant association between the colony counting and the studied independent variable ($p > 0.05$).

### 3.3. Antimicrobial susceptibility test

The results of antimicrobial susceptibility testing of *S. aureus* isolated from raw cow milk, milker's hand, and milking bucket swabs using 13 antimicrobials were shown in Fig 1. A total of 41 *S. aureus* isolates proportionally selected among all isolates of the three towns were tested for antimicrobial susceptibility testing. The highest level of resistance of the isolates was seen for ampicillin (87.8%). On the other hand, the highest numbers of isolates were susceptible to ciprofloxacin (82.9%) and cefotaxime (70.7%).

As shown in Fig 2, out of 41 isolates, 25 (61%) *S. aureus* isolates developed resistance to three or more classes of antimicrobials.

A significantly high percentage of *S. aureus* isolates from milk (75%, 21/28) showed MDR as compared to isolates from hand and milking bucket swab samples ($p = 0.004$) (Table 4).

### 3.4. Risk factors for the contamination of milk with *Staphylococcus aureus*

The univariable logistic regression analysis showed that milk contamination with *S. aureus* was significantly higher in the old-age group lactating cows than in adult and young-age groups lactating cows (crude odds ratio [COR] = 3.91, $p = 0.003$). Similarly, there was a significant difference in the prevalence of *S. aureus* in the milk of the cows with many parities /calvings ($p = 0.001$) and late lactation stages ($p = 0.010$). There was also a significant association between the occurrence of *S. aureus* in the milk and handwashing before and between each milking, dairy house cleaning intervals, and drainage systems in the farms ($p \leq 0.05$). However, there was no significant association between study towns and udder cleaning intervals with the occurrence of *S. aureus* ($p > 0.05$) (Table 5).

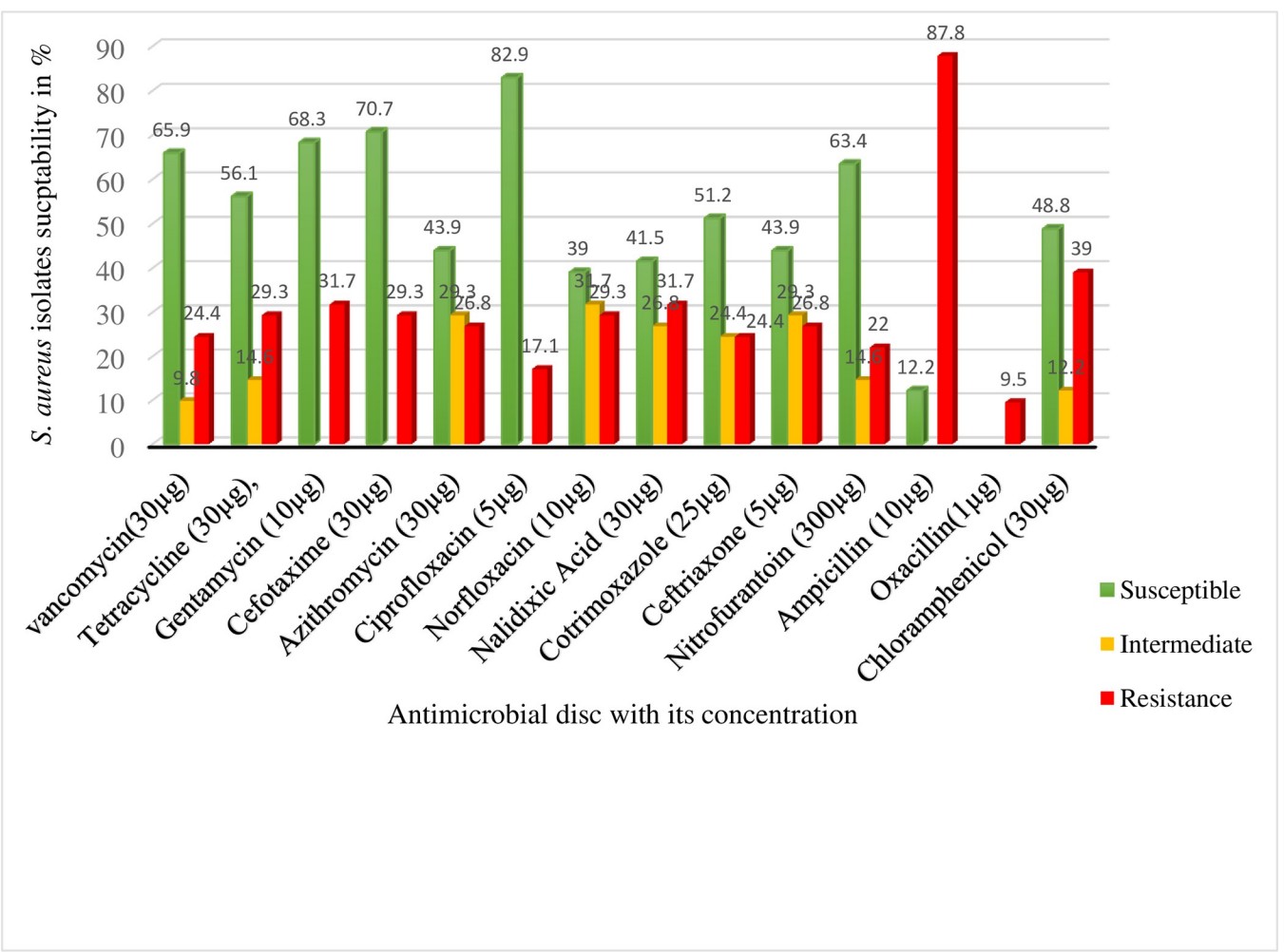

**Fig 1. Results of antimicrobial susceptibility testing of *S. aureus* isolates from milk, hand, and milking bucket swab samples in the study area.**

The variables town and cleaning udder of the cow were excluded from the multivariable logistic regression model because of the univariable p > 0.25. The variables age, parity, lactation stage, hand washing before and between each milking, dairy house cleaning intervals, the drainage systems in the farms as well as farm management were all non-collinear to each other (/r>0.5/) and had univariable p ≤ 0.25, and thus entered into the multivariable logistic regression model (Table 6).

Model selection to identify the best fitting model showed that age, lactation stage of the cows, and dairy house cleaning intervals were the independent predictors of *S. aureus* isolation (Table 7). The data well fitted the model (Hosmer-Lemeshow Chi-square = 19.40, p = 0.431; area under curve (ROC) = 0.786).

## 3.5. KAPs of the farmers on the factors contributing to the occurrence of AMR

The majority of the farmers (51.35%) had moderate knowledge of factors contributing to AMR. Similarly, 49% and 48.65% of the farmers also had a positive attitude and good practices on factors contributing to the occurrence of AMR respectively. The mean knowledge, attitude, and practice score for all farmers were also 7.96, 14.76, and 16.93 respectively (Table 8).

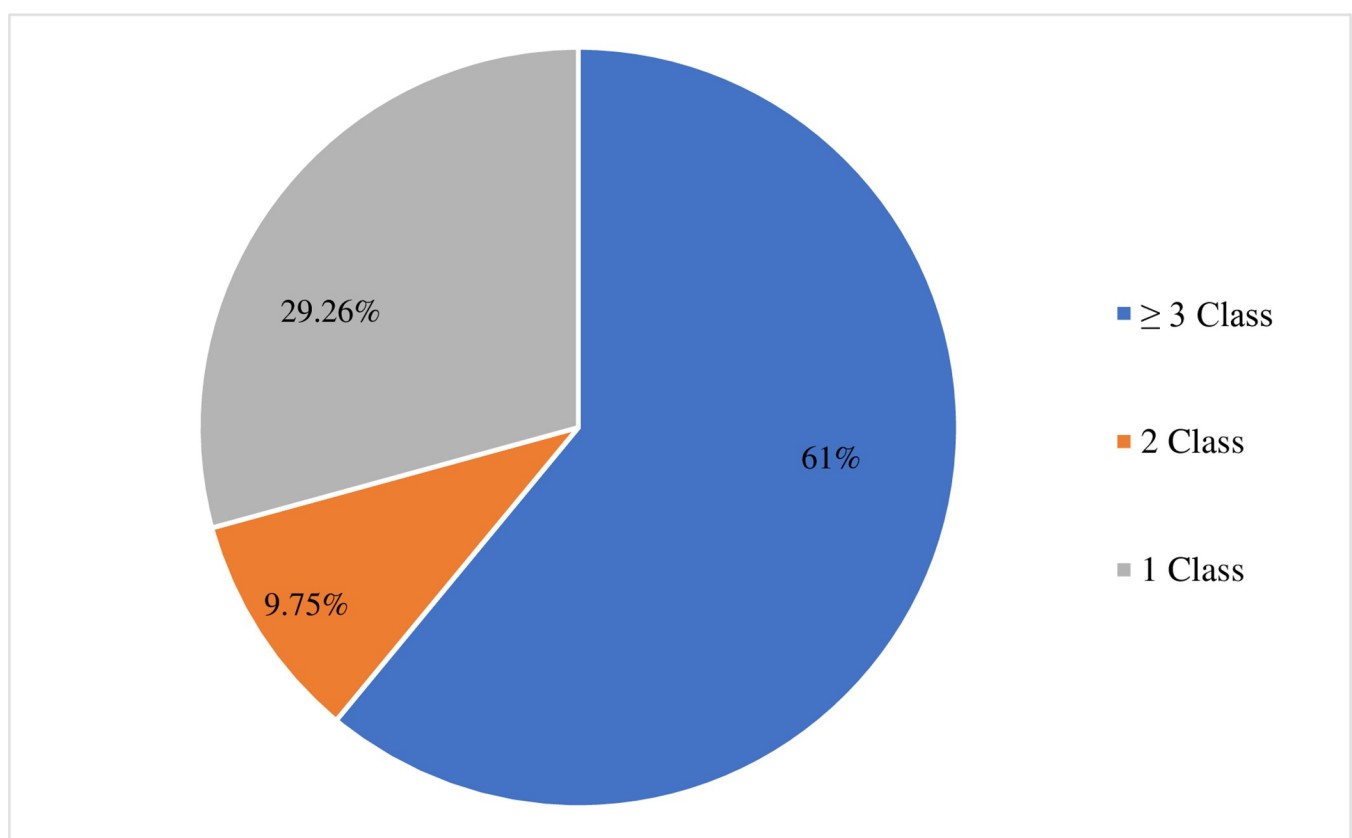

**Fig 2. Percentages of _S. aureus_ isolates resistant to the different antimicrobial classes.**

**3.5.1. Knowledge of farmers on factors causing AMR.**   The response of the farmers to all knowledge questions was summarized in S1 File. More than twenty-one percent (21.6%) of the farmers agreed that treating animals by their own decision without a veterinarian contributes to the occurrence of AMR. Similarly, treating humans by their own decision without the involvement of human health practitioners is another contributing factor to the occurrence of AMR.

**3.5.2. Farmer's attitude towards factors contributing to the occurrence of AMR.**   Nearly half (44.59%) of respondent farmers agreed that inappropriate use of antimicrobials causes AMR. Whereas, 21.62% of the farmers disagreed with the inappropriate use of antimicrobials as one cause of AMR and 43.24% of the farmers agreed on the preference of buying antimicrobials from the pharmacy without a prescription for their animals and family members (S2 File).

**3.5.3. Farmers' practice on factors contributing to the occurrence of AMR.**   The result of the farmers' practice on the factors contributing to the occurrence of AMR was shown in

**Table 4. Number and percentages of resistant _S. aureus_ to antimicrobials in milk, dairy farm workers' hand, and milking bucket swabs.**

| Numbers of resistance antimicrobials classes | Types of samples | | | Total (N, %) | P-value |
|---|---|---|---|---|---|
| | Milk (n, %) | Hand swab (n, %) | Bucket swabs (n, %) | | |
| One | 6 (21.43) | 3 (37.5) | 3(60) | 12 (29.26) | |
| Two | 1 (3.57) | 1 (12.5) | 2(4) | 4 (9.75) | |
| Multidrug resistance (MDR) | 21(75) | 4 (50) | 0 (0.0) | 25 (61) | 0.004 |
| Total | 28 (68.3) | 8 (19.5) | 5 (12.2) | 41(100.0) | |

**Table 5. The result of univariable logistic regression analysis of contributing risk factors for the occurrence and contamination of the milk with _S. aureus_ in the study area.**

| Variable | Category | Tested | Positive (%) | Odds Ratio | 95% CI | P value |
|---|---|---|---|---|---|---|
| Town | Gedo | 54 | 8 (14.8) | 1 | - | - |
| | Ginchi | 67 | 11 (16.4) | 1.13 | 0.42–3.02 | 0.810 |
| | Ejere | 91 | 16 (17.6) | 1.23 | 0.49–3.09 | 0.665 |
| Age | Adult | 93 | 10 (10.75) | 1 | - | - |
| | Young | 69 | 9 (13.04) | 1.25 | 0.48–3.25 | 0.655 |
| | Old | 50 | 16 (32.00) | 3.91 | 1.61–9.47 | 0.003 |
| Parity | Few | 104 | 10 (9.6) | 1 | - | - |
| | Moderate | 86 | 16 (18.6) | 2.15 | 0.92–5.02 | 0.077 |
| | Many | 22 | 9 (40.9) | 6.51 | 2.23–19.00 | 0.001 |
| Lactation stage | Mid | 136 | 15 (11.02) | 1 | - | - |
| | Early | 37 | 9 (24.3) | 2.59 | 1.03–6.53 | 0.043 |
| | Late | 39 | 11 (28.2) | 3.17 | 1.31–7.64 | 0.010 |
| Hand washing before milking | Yes | 105 | 27 (7.62) | 1 | - | - |
| | No | 107 | 8 (25.23) | 4.09 | 1.76–9.50 | 0.001 |
| Hand washing b/n each milking | Yes | 79 | 5 (6.33) | 1 | - | - |
| | No | 133 | 30 (22.56) | 4.31 | 1.60–11.63 | 0.004 |
| Udder of the cow cleaning | Only before milking | 37 | 4 (10.81) | 1 | - | - |
| | Before and after milking | 56 | 9 (16.07) | 1.60 | 0.45–5.56 | 0.477 |
| | No cleaning | 119 | 22 (18.49) | 3.87 | 0.60–5.83 | 0.280 |
| Management | Intensive | 113 | 14 (12.39) | 1 | - | - |
| | Semi-intensive | 99 | 21 (21.21) | 1.90 | 0.91–3.98 | 0.087 |
| Dairy house cleaning interval | Per day | 99 | 7 (7.07) | 1 | - | - |
| | >twice /week | 39 | 5 (12.82) | 1.93 | 0.57–6.50 | 0.287 |
| | twice/week | 74 | 23 (31.08) | 5.93 | 2.38–14.76 | 0.001 |
| Presence of a drainage system | Yes | 137 | 16 (11.68) | 1 | - | - |
| | No | 75 | 19 (25.33) | 2.57 | 0.19–0.81 | 0.012 |

S3 File. More than half of the farmers (51.35%) complete the full course of treatment for themselves and their animals as prescribed.

## 3.6. Association between KAPs toward the factors of AMR in animals and humans

The result of knowledge, attitudes, and practice of the farmers or farm workers on the factors contributing to the occurrence of the AMR revealed a significant association between attitude and knowledge ($\chi^2$ = 50.02; p = 0.001) and practice and knowledge ($\chi^2$ = 38.6; p = 0.001) as indicated in Table 9.

The association of the attitude and practice toward factors contributing to AMR was shown in Table 10. The output of this study also implies that there was a significant association between attitudes and practice (p < 0.05).

## 3.7. The association between socio-demographic variables and KAPs of the farmers on the factors contributing to the AMR

The association of the knowledge, attitude, and practice of the respondents towards the factors contributing to the occurrence of AMR resistance in animals and humans with the socio-demographic variables of the respondents (sex, age, education level, marital status, occupation,

**Table 6. Results of multivariable logistic regression analysis of factors associated with the contamination of the milk with *S. aureus* in the study area.**

| Variable | Category | Adjusted Odds Ratio | 95% CI | P-value |
|---|---|---|---|---|
| Age | Adult | 1 | - | - |
| | Young | 3.36 | 0.79–14.38 | 0.102 |
| | Old | 5.55 | 1.68–18.31 | 0.005 |
| Parity | Few | 1 | - | - |
| | Moderate | 2.29 | 0.56–9.41 | 0.249 |
| | Many | 3.02 | 0.53–17.17 | 0.213 |
| Lactation stage | Mid | 1 | - | - |
| | Early | 2.61 | 0.79–8.63 | 0.115 |
| | Late | 8.7 | 1.53–13.53 | 0.006 |
| Hand washing before milking | Yes | 1 | - | - |
| | No | 1.99 | 0.64–6.13 | 0.233 |
| Hand washing between each milking | Yes | 1 | - | - |
| | No | 3.50 | 0.95–12.83 | 0.059 |
| Management | Intensive | 1 | - | - |
| | Semi-intensive | 1.63 | 0.66–4.00 | 0.288 |
| Dairy house cleaning intervals | Per day | 1 | - | - |
| | >2x/Week | 2.15 | 0.55–8.44 | 0.274 |
| | 2x/Week | 4.78 | 1.56–14.55 | 0.006 |
| Presence of a drainage system | Yes | 1 | - | - |
| | No | 1.18 | 0.46–3.10 | 0.729 |

and religion) was shown in Table 11. There was a significant association between age and knowledge (p = 0.006), age, and practice (p = 0.032). Similarly, the level of education was significantly associated with knowledge (p = 0.001), attitude (p = 0.001), and practice (p = 0.002) (Table 11).

## 4. Discussion

*Staphylococcus aureus* is a prominent opportunistic pathogen that can infect both dairy cattle and humans. It is also the third most commonly reported cause of foodborne diseases worldwide. The high prevalence of *S. aureus* in this study (16.72%) might be linked to conventional hand milking, the lack of regular post-milking teat dip, dairy owners' lack of knowledge about dry cow therapy, and operating procedures along the milk production chains [34].

In the current study, although the difference was not statistically significant, there was a higher prevalence of *S. aureus* from the milker's hand swabs as compared to milk and milking

**Table 7. The best-fitting model for predictors of *S. aureus* isolation from milk in the study area.**

| Variable | Category | Adjusted Odds Ratio | 95% CI | P- value |
|---|---|---|---|---|
| Age | Adult | 1 | - | - |
| | Young | 1.33 | 0.48–3.74 | 0.5853 |
| | Old | 5.54 | 2.01–15.30 | 0.001 |
| Lactation stage | Mid | 1 | - | - |
| | Early | 2.08 | 0.74–5.85 | 0.167 |
| | Late | 3.60 | 1.34–9.75 | 0.012 |
| Dairy house cleaning intervals | Per day | 1 | - | - |
| | >2x/Week | 2.33 | 0.64–8.52 | 0.201 |
| | 2x/Week | 8.70 | 3.14–24.12 | 0.001 |

**Table 8. The distribution of the farmer's knowledge, attitudes, and practice on the factors contributing to the occurrence of AMR in dairy farms and personnel.**

| KAPs | Level | Frequency (%) | Minimum | Maximum | Mean | Standard Deviation (SD) |
|---|---|---|---|---|---|---|
| Knowledge | High | 20 (27.03) | 4 | 12 | 7.96 | 2.10 |
| | Moderate | 38 (51.35) | | | | |
| | Low | 16 (21.62) | | | | |
| Attitude | Positive | 36 (49) | 7 | 21 | 14.76 | 3.23 |
| | Neutral | 23 (31) | | | | |
| | Negative | 15 (20) | | | | |
| Practice | Good | 36 (48.65) | 8 | 24 | 16.93 | 2.93 |
| | Fair | 23 (31.08) | | | | |
| | Poor | 15(20.27) | | | | |

bucket swabs. This might be because 10 to 35% and 20 to 75% of humans are persistent and intermittent carriers of *S. aureus* respectively [14, 16]. The hygienic production of milk is important to ensure the safety of the consumer. However, there is no standard hygienic condition followed by farmers during milk production in Ethiopia [35].

The mean count of *S. aureus* in raw cow milk was 4.3 ± 1.45 log10CFU/ml. The variations in the load of *S. aureus* between towns might be due to the differences in handling practices, sanitary conditions, and operating procedures along the milk production chain [34, 36]. However, the total log10CFU/ml mean count of *S. aureus* in the current study was within the limit of the standard mean count of *S. aureus* in raw milk from healthy cows produced under hygienic conditions which should not be more than 4.7 log10 CFU/ml [37]. Nevertheless, more than 10% of all examined samples had *S. aureus* counts higher than 4.7 log10CFU/ml.

Results of the antimicrobial susceptibility test in the current study indicated that the highest percentages of *S. aureus* isolates were susceptible to ciprofloxacin (82.90%), followed by cefotaxime (70.70%), gentamycin (68.30%), vancomycin (65.90%), nitrofurantoin (63.40%), tetracycline (56.10%), and co-trimoxazole (51.20%). The gradual decrease in susceptibility of *S. aureus* isolates to these antimicrobials might probably be due to the limited uptake of the drug, modification of the drug target, enzymatic inactivation, and active efflux of the drug [38, 39].

The present study showed that *S. aureus* isolates were resistant to ampicillin (87.80%), chloramphenicol (39%), and nalidixic acid (31.70%). The rate of *S. aureus* isolates resistant to ampicillin in this study was higher than other antimicrobials. The probable reasons for the majority of *S. aureus* isolates being resistant to β-lactam antibiotics could be due to the common and prolonged use of the drugs both in human and veterinary practices in Ethiopia [34, 40].

**Table 9. Association between KAPs of the farmer on the factors contributing to the occurrence of AMR in dairy cows and personnel.**

| | | Knowledge | | | | Chi-square ($\chi^2$) | Degree of freedom (df) | P-value |
|---|---|---|---|---|---|---|---|---|
| | Level | High | Moderate | Low | Total | | | |
| | | N (%) | N (%) | N (%) | N (%) | | | |
| Attitude | Positive | 15 (41.67) | 20 (55.56) | 1 (2.78) | 36 (100) | | | |
| | Neutral | 5 (21.74) | 16 (69.57) | 2 (8.70) | 23 (100) | 50.02 | 4 | 0.001 |
| | Negative | 0 | 2 (13.33) | 13 (86.67) | 15(100) | | | |
| Practice | Good | 13 (36.11) | 21 (58.33) | 2 (5.80) | 36 (100) | | | |
| | Fair | 7 (30.43) | 14 (60.87) | 2 (8.70) | 23 (100) | 38.60 | 4 | 0.001 |
| | Poor | 0 | 3 (20.00) | 12 (80.00) | 15 (100) | | | |

**Table 10.  Association between attitudes and practices of the farmer on the factors contributing to the occurrence of AMR in dairy cows and personnel.**

| | Attitude | | | | | Chi-square | df | P-value |
|---|---|---|---|---|---|---|---|---|
| | Level | Positive | Neutral | Negative | Total | | | |
| | | N (%) | N (%) | N (%) | N (%) | | | |
| Practice | Good | 27 (75.00) | 9 (25.00) | 0 | 36 (100) | | | |
| | Fair | 9 (39.35) | 13 (56.52) | 1 (4.35) | 23 (100) | 70.6 | 4 | 0.001 |
| | Poor | 0 (0.00) | 1 (6.67) | 14 (93.33) | 15 (100) | | | |
| Total | | 36 (48.65) | 23 (31.08) | 15 (20.27) | 74 (100) | | | |

The current research identified that *S. aureus* isolates from milk were resistant to oxacillin (9.50%), which is an indicator of the occurrence of MRSA. The occurrences of MRSA in milk are most likely caused by the pervasive use of beta-lactams in dairy cows, poor milk hygiene (sub-standard hygienic procedures practiced by milkers), herd size, and production systems [41, 42]. The MRSA in dairy farms spotlighted the need for increased milkers' awareness concerning safe milk collection and the adoption of good hygienic practices to prevent cross-contamination, as well as the improper prescription and administration of antimicrobials by unauthorized individuals.

The significant and alarming rate of MDR *S. aureus* isolates (61%) found in this investigation may indicate the presence and spread of many virulence genes from resistant strains thus leading to a serious threat to the dairy industry. The high proportion of MDR seen in this study may be due to farmers' improper treatment regime related to unlimited access to antimicrobial drugs, which encourages abuse and increased selection pressure for resistant bacterial strains. Moreover, this is further aggravated by the poor or nonexistent antimicrobial resistance surveillance program [43, 44]. When the level of MDR among *S. aureus* isolates from various sources was evaluated in the current investigation, it became clear that raw milk isolates were substantially more resistant to antimicrobials than those isolated from milker's hand and milking bucket swabs (p = 0.004). This might be because the cows were previously exposed to various antimicrobial drugs or because antimicrobials were used improperly in animals, endangering the effectiveness of current treatments and the capacity to manage infectious diseases in both animals and humans [44, 45].

The high prevalence of *S. aureus* in the milk of old age cows (AOR = 5.55, p = 0.005) compared to milk from adult and young cows might be because older cows have weakened immune systems compared to young and adult cows [46]. Additionally, older cows might have pendulous udders, which are liable for teat and udder injury allowing pathogens to enter the mammary gland [4]. Moreover, as cows' age increases, their teat ends undergo morphological changes (teat dilation), which allow bacteria to enter [47].

**Table 11.  Association between sociodemographic variables and knowledge, attitude, and practice of the farmers or farm workers on the factors contributing to the AMR.**

| Socio-demographic variable | Knowledge | | | Attitude | | | Practice | | |
|---|---|---|---|---|---|---|---|---|---|
| | df | $\chi^2$ | P | df | $\chi^2$ | P | df | $\chi^2$ | P |
| Sex | 2 | 1.9 | 0.388 | 2 | 3.6 | 0.166 | 2 | 5.6 | 0.060 |
| Age | 4 | 14.5 | 0.006 | 4 | 9.5 | 0.051 | 4 | 10.6 | 0.032 |
| Education level | 6 | 33.4 | 0.001 | 6 | 24.3 | 0.001 | 6 | 20.3 | 0.002 |
| Marital status | 2 | 2.7 | 0.263 | 2 | 4.13 | 0.127 | 2 | 1.2 | 0.550 |
| Occupation | 6 | 11.7 | 0.070 | 6 | 7.80 | 0.253 | 6 | 3.80 | 0.704 |
| Religion | 4 | 1.32 | 0.857 | 4 | 3.74 | 0.442 | 4 | 4.50 | 0.290 |

The odds of isolation of *S. aureus* from cows' milk in the late lactation stage (AOR = 8.70; p = 0.006) was significantly higher when compared to cows' in the mid-stage of lactation. *S. aureus* is a contagious pathogen and its' occurrence in cow milk increases as exposure of the cow to this pathogen increases. The variations seen between studies might be attributed to the differences in management practices and the absence of dry cow therapy regimes. When compared to farms that applied cleaning dairy houses daily, the likelihood of isolating *S. aureus* from raw cow milk from those farms cleaning houses twice per week was significantly higher (AOR = 4.78, p = 0.006). This could be the result of unsanitary farm circumstances, such as uncleaned bedding that exposes teat ends and makes it easier for *S. aureus* to infect the cow's udder because it is so ubiquitous [48, 49].

According to the findings of the current KAP study, farmers' knowledge of the risks associated with AMR and the contributing factors to AMR in dairy cows and personnel was insufficient.The present study found that 40.54% of respondents agree with the knowledge questions that over or under-dose use of antimicrobials in animals and humans results in the occurrence of AMR which was consistent with research done in Addis Ababa, Ethiopia [50], where 36.6% knew that using antimicrobials excessively or insufficiently causes AMR to occur in both humans and animals. However, the finding of this study was lower than the report from the six Vietnamese provinces, Vietnam [51] and Rupandehi district, Nepal [52] in which 95% of the livestock and aquaculture producers and 70% of respondents on antibiotic use among community members agreed that proper use of antimicrobials could help to reduce the risk of AMR respectively. Sixteen percent of the respondents disagreed that the frequent use of the same antibiotic in humans and animals might cause AMR, which was in contrast with the report from Ilala, Kibaha, and Kilosa, Tanzania [53] in which 55.60%, 46.20%, and 34.10% of the respondent agreed with frequent use antimicrobials had a risk for AMR respectively. In the current study, 49% of the respondents had a positive attitude toward the factors contributing to AMR. In contrast to this, 14.71% of animal producers in Oromia Zone northeastern Ethiopia had a positive attitude regarding antimicrobial use and antimicrobial resistance [54]. In the present study, 44.59% of the respondents agreed that inappropriate use of antimicrobials both in humans and animals can causes AMR; which was higher than the report from Kemissie town, Ethiopia, in which 17.70% of the respondent believed that misuse of antimicrobials results in AMR [55]. Similarly, the current study showed that 33.78% of the respondents believed missing one or two doses does not alter the effectiveness of antimicrobials. Unlike the current findings, 77% of the respondents in Bangladesh agreed that missing the doses of the antimicrobial result in AMR [56]. The probable reason for the difference among these studies might be that there was an awareness creation variation among the communities. In the present study, 48.65% of the respondents had good practice toward the factors contributing to the occurrence of AMR. The report of the current study on practice showed that 17.50% of the farmers agreed with keeping leftover antimicrobials for future use, which was in close agreement with the report from the Amhara and Oromia regions, Ethiopia (21.70%) [57]. The present study showed that there was a significant association (p = 0.001) between the farmer's knowledge with attitude and practice toward the factors contributing to AMR. According to this study the attitude, practice, and knowledge of the respondents on the contributing factors of AMR were positively correlated. This showed that as knowledge of the respondents' toward the factor contributing to AMR increases, their attitude and practice also increase.

The result of the current study indicated that socio-demographic factors of the farmers like age and educational level had a significant association (p<0.05) with knowledge, attitudes, and practice toward factors contributing to the occurrence of AMR, which was in close agreement with a study conducted on KAPs of livestock farmers on AMR and AMU in five African countries (Ghana, Kenya, Tanzania, Zambia, and Zimbabwe) [58] where socio-demographic factors

like educational level and KAPs of the livestock farmers across the countries were positively correlated. Unlike the present study, the report from Eastern Turkey [59] showed that demographic factors such as age and educational level did not play a role in the factors contributing to the AMR.

The development of AMR although related to several factors, in this study, could also be related to the inadequate knowledge, attitude, and practice of farmers. This might have potential negative repercussions on food security, food safety, contamination of the environment, and considerable economic losses.

The current study conducted on exotic and crossbred dairy farms indicates that *S. aureus* is a significant pathogen for dairy farmers by negatively affecting milk quantity and quality.The findings also have significant health and public policy ramifications. Future studies should concentrate on conducting comparable experiments on local zebu cows so that the results may be compared.

The study's limitations include the inability to carry out molecular investigations, such as genotypic analysis to precisely identify *Staphylococci* species and detection of genes responsible for antimicrobial resistance and enterotoxins, the inability to perform antimicrobial susceptibility testing for all isolated organisms, and colony counting for all milk samples due to the lack of funding. Furthermore, due to the small number of respondents used to study KAP, generalizing the findings outside the study participants should be done with caution as it may not reflect the true KAP in other areas/regions.

## 5. Conclusions

The study revealed a high prevalence of *Staphylococcus aureus* in milker's hand swabs followed by raw cow milk and milking bucket swabs. The overall mean of *S. aureus* count from raw cow milk in the study area was within the international acceptable range. A large percentage of *S. aureus* isolates were susceptible to ciprofloxacin but resistant to various antimicrobial agents, particularly ampicillin. Moreover, the majority of the isolates tested were resistant to three or more antimicrobial classes, asserting the presence of MDR. Age, lactation stages, and dairy house cleaning intervals were crucial predictors for the occurrence of *S. aureus* in raw cow milk. The majority of the farmers in this study area had a misunderstanding, insufficient knowledge, and practice as well as negative attitudes toward AMR and its contributing factors. Therefore, improved hygienic practices, continuous awareness creation training about AMU and AMR, and a surveillance program of AMR are recommended.

## Supporting information

**S1 File. Farmers' knowledge of factors causing AMR.**
(DOCX)

**S2 File. The response of the farmers to attitude questions on the factors causing AMR.**
(DOCX)

**S3 File. The response of the farmers to the practice questions on the factors causing AMR.**
(DOCX)

**S4 File.**
(XLSX)

## Author Contributions

**Conceptualization:** Endrias Zewdu Geberemedhin.

**Data curation:** Milsan Getu Banu.

**Formal analysis:** Milsan Getu Banu, Endrias Zewdu Geberemedhin.

**Funding acquisition:** Endrias Zewdu Geberemedhin.

**Investigation:** Milsan Getu Banu.

**Methodology:** Milsan Getu Banu, Endrias Zewdu Geberemedhin.

**Project administration:** Endrias Zewdu Geberemedhin.

**Resources:** Endrias Zewdu Geberemedhin.

**Software:** Endrias Zewdu Geberemedhin.

**Supervision:** Endrias Zewdu Geberemedhin.

**Validation:** Endrias Zewdu Geberemedhin.

**Writing – original draft:** Milsan Getu Banu.

**Writing – review & editing:** Endrias Zewdu Geberemedhin.

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
