## [Decision Letter · Decision Letter 0]

9 Aug 2022

PONE-D-22-19890Occurrence and antimicrobial susceptibility of Staphylococcus aureus in dairy farms and personnel in selected towns of West Shewa Zone, Oromia, EthiopiaPLOS ONE

Dear Dr. Zewdu,

Thank you for submitting your manuscript to PLOS ONE. After careful consideration, we feel that it has merit but does not fully meet PLOS ONE’s publication criteria as it currently stands. Therefore, we invite you to submit a revised version of the manuscript that addresses the points raised during the review process. Please ensure that you attend to all concerns raised in the reviewers' comments before your manuscript can be processed further.

We look forward to receiving your revised manuscript.

Kind regards,

Ismail Ayoade Odetokun, DVM, Ph.D.

Academic Editor

PLOS ONE

Journal Requirements:

4. We note you have included a table to which you do not refer in the text of your manuscript. Please ensure that you refer to Table 3 in your text; if accepted, production will need this reference to link the reader to the Table.

Reviewers' comments:

Reviewer's Responses to Questions

**Comments to the Author**

1. Is the manuscript technically sound, and do the data support the conclusions?

Reviewer #1: Yes

Reviewer #2: Partly

Reviewer #3: Yes

Reviewer #4: Yes

2. Has the statistical analysis been performed appropriately and rigorously? 

Reviewer #1: Yes

Reviewer #2: Yes

Reviewer #3: Yes

Reviewer #4: Yes

3. Have the authors made all data underlying the findings in their manuscript fully available?

Reviewer #1: Yes

Reviewer #2: Yes

Reviewer #3: Yes

Reviewer #4: Yes

4. Is the manuscript presented in an intelligible fashion and written in standard English?

Reviewer #1: Yes

Reviewer #2: Yes

Reviewer #3: Yes

Reviewer #4: Yes

5. Review Comments to the Author

Reviewer #1: The manuscript presents the results of a study on the assessment of the presence of Staphylococcus aureus in dairy farms including milk and environmental samples. The results are accompanied with a detailed survey on the hygiene conditions of the farms and training/knowledge of the management providing a good picture of the environment and sector considered in the study and offers consideration for the mitigating of risk factors in the environment that affect human health and disease.

The experimental design and results are well presented, however, redundancies on some sections must be addressed along with including in the discussion a few points to support the reasons why were not presented other detailed data:

- method used for the identification. An ISO method is available for the horizontal method for the enumeration of coagulase-positive staphylococci by counting the colonies obtained on a solid medium (Baird-Parker agar medium). Blood agar medium is not the ideal choice for the detection of S.aureus and other coagulase positive staphylococci.

- Detection of staphylococcal enterotoxins (SEs). Even though SEs are mentioned on the main text, there is no detection either in the milk or as gene presence in the considered strains. This could be addressed through PCR tests for the detection of at least the most known SEs.

-material and methods section - L115-128 please reduce the wording and the information, if necessary to support discussion, could be presented in a table instead (coordinates, msl, rainfall and so on).

-Geographical and meteorological data. The information provided in the m&m section is interesting, however, there is no contextually consideration in the discussion/ conclusion. If not relevant these informations can be removed.

Reviewer #2: I am quite concern with the antibiotic susceptibility testing in this manuscript, I did not see the author performed any testing that can indicate whether the S. aureus strains isolated are MRSA or MSSA ? when we work with S. aureus strain, this testing should be performed because its really related with pathogenesis of this bacteria and off course with its antimicrobial susceptibility profile. Author(s) can easily identify this by testing cefoxitin or oxacillin disc. I do not think ampicillin is necessary to test in the case of S. aureus.

Please clarify which standard reference the author(s) used on their antibiotic susceptibility testing? is it CLSI or EUCAST?

The number of table in the main manuscript a bit too much. I suggest the author put some of these table in the supplementary data so the manuscript will not be too crowded

Reviewer #3: This cross-sectional study by Endrias, Z.G., and Milsan, G.B., 2022 was conducted in selected towns of the West Shewa Zone, Oromia, Ethiopia from December 2020 to April 2021 with the aim of:

1.estimating the occurrence and load of S. aureus in raw cows’ milk,

2.determining the antimicrobial susceptibility patterns of the S. aureus isolates, and

3.assessing knowledge, attitude, and practice of the farmers on factors of antimicrobial resistance.

Dependent (outcome) variables: occurrence and load of S. aureus in raw cows’ milk, antimicrobial susceptibility patters of S. aureus isolates, and KAP of farmers on factors of AMR.

Independent (predictors) variables: predictors of S. aureus isolation include age of the cow, lactation stage and dairy house cleaning intervals. Predictors of KAP of the farmers include their age, sex, educational level, marital status, education, and religion. Predictors of AMR to S. aureus isolates include over or under dose use of antimicrobials, frequent use of the same antibiotic, farmers attitude towards antibiotic use and AMR, missing the doses of the antimicrobial agent, and awareness creation variation among the communities.

A total of 311 samples from raw cows’ milk (212), milkers’ hand (44), and milking bucket (55) swabs were collected and tested. The disc diffusion method was used to test antimicrobial susceptibility of the 38 isolates. Questionnaire survey was conducted to assess the factors of milk contamination with S. aureus and antimicrobial resistance. The Chi-square test, one-way analysis of variance, and logistic regression analysis were used for data analyses. This work was reviewed and approved by the Ambo University Research and Ethical Committee.

The result indicated that 16.72% (52/311) (95% CI: 12.75-21.34%) of the samples were positive for S. aureus. The occurrence of S. aureus was 22.73%, 16.51%, and 12.73% in milkers’ hand swabs, cow milk, and milking bucket swabs, respectively. The mean count of S. aureus from raw cows’ milk was 4.3± 1.45 log10 CFU/ml. On AMR, about 88% of S aureus isolates were resistant to ampicillin while 82.9% and 70.7% of the isolates were susceptible to ciprofloxacin and cefotaxime respectively. Majority of the S. aureus isolates (61%) showed multi-drug resistance. The odds of S. aureus isolation from the milk of cows were significantly high in older cows (Adjusted Odds Ratio [AOR]: 5.54; p= 0.001), in late lactation stages (AOR: 3.6; p = 0.012), and in farms where house cleaning was done twice per week (AOR:

49 8.7; p = 0.001). Lastly, high percentage of farmers had insufficient knowledge, attitude, and practice (KAP) about the factors contributing for antimicrobial resistance.

The authors made all data underlying the findings fully available. The data was tested for representativeness, analyzed using descriptive and inferential statistics which were rigorous and appropriate.

Discussions of the results were robust, citing similar studies conducted both within and outside Ethiopia.

Conclusions are in line with the findings

Writing quality and clarity: Satisfactory

Other observations:

1.Limitations of the study: The authors did well to mention the limitations of the study

2.Inclusion/exclusion criteria clearly explained as a separate subtopic.

However, I am not sure if the authors explained why the study excluded the local lactating cows (Zebu) and opted for healthy cross breed (Zebu with Jersey and Holstein-Friesians) lactating cows. In my view, this could constitute a source of bias for the study.

As a way forward, the authors can consider conducting similar study with the local lactating cows and compare the outcomes with this one. In this way the sample size will increase, and generalizability of the results can be enhanced.

Reviewer #4: PONE-D-22-19890 Comments

Zwedu Zwedu et al., investigated the occurrence and antimicrobial susceptibility of Staphylococcus aureus in dairy farms and personnel in selected towns of West Shewa Zone, Oromia, Ethiopia. The manuscript is well-written, and results are presented clearly and precisely.

While the study and the findings are interesting, there are a few ways in which the manuscript can be improved.

Abstract:

Line 1: Staphylococcus aureus is one of the foodborne causing bacterial…..?

Please complete the sentence with either bacterial pathogen or bacteria.

There are some incomplete sentences throughout the manuscript, please make the necessary changes (e.g., lines 59-61; 65; etc.)

Introduction:

Line 59-61: Staphylococcus aureus is an opportunistic bacterial pathogen with the capability to persist and multiply in a variety of environments; predominantly incriminated as one of the major causes of foodborne diseases

Line 65: and public health what? Please complete the sentence.

Methods:

What is the rationale for choosing the towns used in the study? Why? Are those the highest milk-producing/marketing towns? Why three towns?

Sample collection from milkers’ hands: What period? Before or immediately/several hours after milking? Before or after washing hands? Why? Was the method consistent throughout all sampling? Would that add variability to the results?

Sample collection from milking buckets: When? How? Same procedure/spot for all samples?

Cows: samples were consistently collected at the same time/period and from similar lactation stages in all cows in all the sample towns?

Discussion:

Majority of the discussion centered around comparison of numbers/statistics between regions in and out of Ethiopia…it’s pretty difficult to keep up with all these numbers. All these may be part of the discussion, but the discussion section should mainly discuss the significance of the findings and what the results mean for the regions where the study was conducted…not just observational and reporting statistics but discussion of the relevance and significance of the findings and the study as a whole.

Line 499-506: There’s too many speculations as to why S. aureus is more prevalent in older cows than others…speculations should be limited unless there’s a direct association or probable explanation from the findings of the current study.

Also, comparison of findings from this study with data from regions with entirely socioeconomic, cultural, and political differences seems far off, and makes the discussion bounce back and forth as to why these comparisons were made. Why? What’s the rationale of comparing data between regions?

6. PLOS authors have the option to publish the peer review history of their article (what does this mean?). If published, this will include your full peer review and any attached files.

Reviewer #1: No

Reviewer #2: No

Reviewer #3: **Yes: **Haruna Ismaila Adamu, MBBS; MPH; PhD

Reviewer #4: No

---

## [Author Response · Author response to Decision Letter 0]

26 Aug 2022

Responses to editor:

1. Please ensure that your manuscript meets PLOS ONE's style requirements, including those for file naming 

Corrected as per the journals’ requirements

The study participants for the questionnaire survey were briefed about the study and gave informed oral consent before the collection of the data.

In the study we didn’t involve minors

Captions for both figures have been included in the manuscript

4. We note you have included a table to which you do not refer in the text of your manuscript. Please ensure that you refer to Table 3 in your text; if accepted, production will need this reference to link the reader to the Table.

Table 4 was cited in the text of the revised version

Responses to queries of Reviewer #1

1. The experimental design and results are well presented, however, redundancies on some sections must be addressed along with including in the discussion a few points to support the reasons why were not presented other detailed data:

Redundancies were checked and addressed.

In the discussion explanations /reasoning’s were added

2. Method used for the identification. An ISO method is available for the horizontal method for the enumeration of coagulase-positive staphylococci by counting the colonies obtained on a solid medium (Baird-Parker agar medium). Blood agar medium is not the ideal choice for the detection of S. aureus and other coagulase positive Staphylococci.

We used the Baird-Parker agar medium for colony counting. But, blood agar medium was used only for the detection of presence or absence of staphylococci in the samples and not for confirmatory purpose.

3. Detection of staphylococcal enterotoxins (SEs). Even though SEs are mentioned on the main text, there is no detection either in the milk or as gene presence in the considered strains. This could be addressed through PCR tests for the detection of at least the most known SEs.

We put the consequence staphylococcal enterotoxins (SEs) for general information. But, due to limitation of resource for molecular testing, we didn’t perform the test (it was discussed as limitation of the study)

4. Material and methods section - L115-128 please reduce the wording and the information, if necessary to support discussion, could be presented in a table instead (coordinates, msl, rainfall and so on).

It is accepted and the sentence was re-written

5. Geographical and meteorological data. The information provided in the m&m section is interesting, however, there is no contextually consideration in the discussion/ conclusion. If not relevant these information can be removed.

It is accepted and re-written 

Responses to queries of Reviewer #2

1. I am quite concern with the antibiotic susceptibility testing in this manuscript, I did not see the author performed any testing that can indicate whether the S. aureus strains isolated are MRSA or MSSA ? when we work with S. aureus strain, this testing should be performed because its really related with pathogenesis of this bacteria and off course with its antimicrobial susceptibility profile. Author(s) can easily identify this by testing cefoxitin or oxacillin disc. I do not think ampicillin is necessary to test in the case of S. aureus.

The comment is valid and well accepted. We did MRSA or MSSA for 21 samples by using oxacillin disc due to the inadequate number of discs we had. This result was presented in the result section as well discussed in the revised version.

In this research, we tested for ampicillin because this antibiotic is commonly used in the community of our study area.

2. Please clarify which standard reference the author(s) used on their antibiotic susceptibility testing? is it CLSI or EUCAST? The standard we used for the antibiotic testing was CLSI, 2020.

3. The number of table in the main manuscript a bit too much. I suggest the author put some of these table in the supplementary data so the manuscript will not be too crowded. 

Accepted and corrected as suggested (Table 9, 10 and 11 were changed into supplementary file 1, 2 and 3 respectively.

Responses for queries of reviewer#3

1. Inclusion/exclusion criteria clearly explained as a separate subtopic. However, I am not sure if the authors explained why the study excluded the local lactating cows (Zebu) and opted for healthy cross breed (Zebu with Jersey and Holstein-Friesians) lactating cows. In my view, this could constitute a source of bias for the study.

In the three purposively selected towns the dairy farms are mainly managed under intensive or semi-intensive management system and the breeds farmers keep for milk purpose are exotic and their crossbred (keeping zebu cows for milk purpose in the towns was not common since they are poor in milk production). However, the issue of local lactating cows (zebu) was raised in the discussion as future areas of research

2. As a way forward, the authors can consider conducting similar study with the local lactating cows and compare the outcomes with this one. In this way the sample size will increase, and generalizability of the results can be enhanced. 

i. Addressed in the previous response (line 522-524).

Responses to queries of reviewer#4

1. Abstract:

Line 1: Staphylococcus aureus is one of the foodborne causing bacterial…..?

Please complete the sentence with either bacterial pathogen or bacteria.

There are some incomplete sentences throughout the manuscript, please make the necessary changes (e.g., lines 59-61; 65; etc.).

It is accepted and corrected 

2. Introduction:

Line 59-61: Staphylococcus aureus is an opportunistic bacterial pathogen with the capability to persist and multiply in a variety of environments; predominantly incriminated as one of the major causes of foodborne diseases

Line 65: and public health what? Please complete the sentence.

It is accepted and corrected

3. Methods

i. What is the rationale for choosing the towns used in the study? Why? Are those the highest milk-producing/marketing towns? Why three towns?

The three towns were purposively selected by considering the potential for dairy production, logistics and the absence of previous studies on the topic (lines 117-118).

ii. Sample collection from milkers’ hands: What period? Before or immediately/several hours after milking? Before or after washing hands? Why? Was the method consistent throughout all sampling? Would that add variability to the results? 

The milker’s hand swabs were collected consistently throughout the study immediately before milking because normally human being are the carriers of this bacteria and cross checked with cow’s milk and add variability on the results. Not only the cow but also the milkers could also be the source of contamination of raw milk, thus it might potentially add variability to the result.

iii. Sample collection from milking buckets: When? How? Same procedure/spot for all samples? 

The sample collection from milking buckets (inner wall) was done consistently before milking for all swabs. The procedures were explained under section 2.6 (Sample collection and transportations)

iv. Cows: samples were consistently collected at the same time/period and from similar lactation stages in all cows in all the sample towns?

The samples were collected from cows of different lactation stages with the same procedure, not at the same time /period.

4.Discussion

i. Majority of the discussion centered around comparison of numbers/statistics between regions in and out of Ethiopia…it’s pretty difficult to keep up with all these numbers. All these may be part of the discussion, but the discussion section should mainly discuss the significance of the findings and what the results mean for the regions where the study was conducted…not just observational and reporting statistics but discussion of the relevance and significance of the findings and the study as a whole.

Accepted and addressed in detail in revised version

ii. Line 499-506: There’s too many speculations as to why S. aureus is more prevalent in older cows than others…speculations should be limited unless there’s a direct association or probable explanation from the findings of the current study.

In the revised version the speculations were minimized and few speculation along explanations were included.

iii. Also, comparison of findings from this study with data from regions with entirely socioeconomic, cultural, and political differences seems far off, and makes the discussion bounce back and forth as to why these comparisons were made. Why? What’s the rationale of comparing data between regions?

Although the intention of comparing data between the regions of this study were to imply different risk factors, method, laboratory tests, and variability in prevalence and farmers KAPs on factors of AMR in different geographical areas, we agreed to minimize the text of discussion by eliminating comparisons. Therefore, the discussion was modified as be the suggestions given.

5.While revising your submission, please upload your figure files to the Preflight Analysis and Conversion Engine (PACE) digital diagnostic.

Done as per the requirement

---

## [Decision Letter · Decision Letter 1]

27 Oct 2022

PONE-D-22-19890R1Occurrence and antimicrobial susceptibility of Staphylococcus aureus in dairy farms and personnel in selected towns of West Shewa Zone, Oromia, EthiopiaPLOS ONE

Dear Dr. Zewdu,

Thank you for submitting your manuscript to PLOS ONE. After careful consideration, we feel that it has merit but does not fully meet PLOS ONE’s publication criteria as it currently stands. Therefore, we invite you to submit a revised version of the manuscript that addresses the points raised during the review process.

We look forward to receiving your revised manuscript.

Kind regards,

Ismail Ayoade Odetokun, DVM, Ph.D.

Academic Editor

PLOS ONE

Journal Requirements:

Additional Editor Comments:

Though all comments have been addressed, the manuscript needs a thorough proofreading (punctuations and spacing between words, etc.) prior to acceptance for publication.

Reviewers' comments:

Reviewer's Responses to Questions

**Comments to the Author**

1. If the authors have adequately addressed your comments raised in a previous round of review and you feel that this manuscript is now acceptable for publication, you may indicate that here to bypass the “Comments to the Author” section, enter your conflict of interest statement in the “Confidential to Editor” section, and submit your "Accept" recommendation.

Reviewer #1: All comments have been addressed

Reviewer #3: All comments have been addressed

Reviewer #4: All comments have been addressed

2. Is the manuscript technically sound, and do the data support the conclusions?

Reviewer #1: Yes

Reviewer #3: Yes

Reviewer #4: Partly

3. Has the statistical analysis been performed appropriately and rigorously? 

Reviewer #1: N/A

Reviewer #3: Yes

Reviewer #4: Yes

4. Have the authors made all data underlying the findings in their manuscript fully available?

Reviewer #1: Yes

Reviewer #3: Yes

Reviewer #4: Yes

5. Is the manuscript presented in an intelligible fashion and written in standard English?

Reviewer #1: Yes

Reviewer #3: Yes

Reviewer #4: Yes

6. Review Comments to the Author

Reviewer #1: (No Response)

Reviewer #3: All my concerns in the previous review have been adequately addressed, including suggestions for future research studies

Reviewer #4: (No Response)

7. PLOS authors have the option to publish the peer review history of their article (what does this mean?). If published, this will include your full peer review and any attached files.

Reviewer #1: No

Reviewer #3: **Yes: **Haruna Ismaila ADAMU, MBBS; MPH; PhD

Reviewer #4: No

---

## [Author Response · Author response to Decision Letter 1]

2 Nov 2022

The following is a point by point responses given to the queries raised during the second round review process.

First, we would like to extend our sincere thanks and appreciations once again to all the reviewers and editors for taking their time in suggesting important comments towards improving the quality of our manuscript. In the table below we have indicated our response to each queries and accordingly incorporated the revisions in the manuscript by highlighting the changes in red for easy tracing. The language aspect of the manuscript was also substantially revised.

Response to editors’ comments:

Queries:Please review your reference list to ensure that it is complete and correct. If you have cited papers that have been retracted, please include the rationale for doing so in the manuscript text, or remove these references and replace them with relevant current references. Any changes to the reference list should be mentioned in the rebuttal letter that accompanies your revised manuscript. If you need to cite a retracted article, indicate the article’s retracted status in the References list and also include a citation and full reference for the retraction notice.

Response: To the best of our knowledge we have revised the reference section to fit to the journal style by correcting all errors. No retracted paper was cited

Queries /additional editor's comments: Though all comments have been addressed, the manuscript needs a thorough proofreading (punctuations and spacing between words, etc.) prior to acceptance for publication.

Response: Thank you once again for the comments. In this revised version we tried to correct such errors of spacing, punctuations, typographical, grammar, and the like.

---

## [Editor Report · Decision Letter 2]

4 Nov 2022

Occurrence and antimicrobial susceptibility of Staphylococcus aureus in dairy farms and personnel in selected towns of West Shewa Zone, Oromia, Ethiopia

PONE-D-22-19890R2

Dear Dr. Zewdu,

We’re pleased to inform you that your manuscript has been judged scientifically suitable for publication and will be formally accepted for publication once it meets all outstanding technical requirements.

Kind regards,

Ismail Ayoade Odetokun, DVM, Ph.D.

Academic Editor

PLOS ONE
---

## [Editor Report · Acceptance letter]

8 Nov 2022

PONE-D-22-19890R2 

Occurrence and antimicrobial susceptibility of *Staphylococcus aureus* in dairy farms and personnel in selected towns of West Shewa Zone, Oromia, Ethiopia 

Dear Dr. Zewdu:

I'm pleased to inform you that your manuscript has been deemed suitable for publication in PLOS ONE. Congratulations! Your manuscript is now with our production department. 

Kind regards, 

on behalf of

Dr. Ismail Ayoade Odetokun 

Academic Editor

PLOS ONE